# Interpersonal Dynamics of Authentic Leadership: Effects on Support Perception and Workplace Procrastination

Sergio Edú-Valsania [1], Ana Laguía [2,*] and Juan Antonio Moriano [2]

1   Department of Social Sciences, Universidad Europea Miguel de Cervantes, 47012 Valladolid, Spain; sedu@uemc.es
2   Department of Social and Organizational Psychology, Universidad Nacional de Educación a Distancia (UNED), 28040 Madrid, Spain; jamoriano@psi.uned.es
*   Correspondence: aglaguia@psi.uned.es

**Abstract:** (1) Background: Workplace procrastination leads to missed deadlines and financial losses, necessitating a deeper understanding of its risk factors and inhibitors for effective interventions. This study aims to bridge the significant gap in the literature regarding the effects of Authentic Leadership (AL) on workplace procrastination behaviors, including soldiering and cyberslacking. AL, as a positive leadership style, is proposed as a key factor in mitigating procrastination by fostering a supportive work environment. Specifically, this research examines how AL impacts procrastination through two psychosocial risk factors—lack of supervisor support and lack of workgroup support—which are hypothesized to mediate this relationship. (2) Methods: Data were collected from 738 employees (62.9% women) who completed a survey. Partial least squares structural equation modeling was conducted to explore the direct relationship between AL and procrastination, and indirect relationships through support. (3) Results: The findings indicate that AL negatively impacts procrastination behaviors, with stronger effects on soldiering compared to cyberslacking. AL is also negatively associated with perceptions of a lack of support from both leaders and workgroups, with a stronger influence on leader support. Both lack of leader and workgroup support significantly predict soldiering but not cyberslacking. (4) Conclusions: This study highlights AL's potential to mitigate workplace procrastination by reducing perceptions of insufficient support. Organizations should focus on AL training to promote leader authenticity and supportiveness while fostering strong support networks within workgroups to enhance productivity and reduce procrastination behaviors. These findings also contribute to understanding AL's role in addressing workplace counterproductive behaviors.

**Keywords:** authentic leadership; procrastination; soldiering; cyberslacking; leader support; workgroup support

## 1. Introduction

The Society for Industrial and Organizational Psychology (Howald et al., 2018) underscored the critical importance of managing and preventing deviant and counterproductive workplace behaviors due to their significant economic costs to organizations. Deviant or counterproductive behaviors refer to employees' voluntary behaviors that are consciously or unconsciously aimed at damaging individual or organizational performance (Thrasher et al., 2020). Current research increasingly focuses on one of these behaviors, namely, workplace procrastination, identified as a major cost factor due to decreased productivity

(Bellini et al., 2022; Metin et al., 2020; Sirois, 2023; Unda-López et al., 2022). These behaviors, prevalent across many organizations, lead to missed deadlines and represent a high associated economic cost (Paulsen, 2015).

Despite growing awareness of workplace procrastination, existing studies primarily concentrate on its consequences, such as psychological discomfort, psychological detachment, stress, exhaustion at work, and negative emotions across various populations and contexts (Bellini et al., 2022; Sirois, 2023). This highlights the need for further research to explore not only these outcomes but also the antecedents and protective factors that contribute to these behaviors. In particular, understanding the role of leadership and support systems in shaping these negative emotions and behaviors may be essential for designing effective interventions that can mitigate workplace procrastination and its associated outcomes.

Leadership has been identified as a facilitator or inhibitor of employee procrastination (He et al., 2023; Liao et al., 2023; Lin, 2018; Singh et al., 2021). Considering this critical role, examining positive leadership styles that can foster an environment conducive to reducing procrastination is particularly important. One of the most widely studied theoretical frameworks related to positive leadership is Authentic Leadership (AL), which has received considerable attention from many scholars for its relationships with a wide variety of organizational and employee-related outcomes (Luthans & Avolio, 2003). Authentic leaders are characterized as transparent, trustworthy, humble, ethical, and inclusive. They promote positive emotions, attitudes, and behaviors in employees, such as organizational citizenship behaviors (OCBs), knowledge-sharing behaviors, intrapreneurial behavior, organizational and group identification, and a climate for innovation, among others (Banks et al., 2016; Edú-Valsania et al., 2012, 2016a, 2016b).

Furthermore, the literature identifies two critical elements of interpersonal dynamics within leadership that directly impact workplace procrastination: the perceived lack of support from leaders and coworkers. When employees experience this absence of support, they may face anxiety, stress, and other negative emotions (Farfán et al., 2019; García-Buades et al., 2024), which are closely linked to procrastination (Metin et al., 2016; Sirois, 2023). Indeed, insufficient support from supervisors and coworkers is considered a major source of workplace stress (Spielberger et al., 2003). Thus, in this study, we will focus on two psychosocial risk factors—lack of supervisor support and lack of workgroup support—as potential facilitators of employee procrastination.

Therefore, this study seeks to address the gap in the literature concerning the role of AL in influencing workplace procrastination and the levels of support employees perceive from their leaders and coworkers. To date, the connection between AL and both employee procrastination and perceived support has been largely unexplored. Effective leadership should not only foster positive outcomes but also proactively prevent situations that may lead to counterproductive employee behaviors. Consequently, the objectives of this research are as follows:

1. To explore the relationship between AL and employee procrastination behaviors at work.
2. To examine the relationship between AL and perceptions of lack of support from the leader and the workgroup.
3. To analyze the potential role of lack of supervisor and workgroup support in employees' procrastination at work.
4. To investigate the possible mediating effects of lack of supervisor and workgroup support on the relationship between AL and employees' procrastination at work.

*1.1. Theoretical Background and Hypotheses*

1.1.1. Procrastination at Work

Procrastination, in general, can be understood as a form of self-regulatory failure characterized by an irrational delay of tasks despite potentially negative consequences for the procrastinators (Prem et al., 2018), which is often preceded by psychological discomfort, detachment, stress, exhaustion, and negative emotions (Bellini et al., 2022; Sirois, 2023; Steinert et al., 2021). The phenomenon of procrastination is not exclusive to the work environment. In fact, the vast majority of studies on procrastination have primarily been conducted in academic and clinical settings (Yan & Zhang, 2022). Specifically for work environments, Metin et al. (2016, p. 255) defined procrastination at work as "the delay of work-related action by intentionally engaging (behaviorally or cognitively) in nonwork-related actions, with no intention of harming the employer, employee, workplace or client" and highlighted that individuals are aware of the tasks that need to be completed but lack the self-motivation to execute them within a certain timeframe. These authors proposed a two-factor model of procrastination at work, which has been validated in several countries (Metin et al., 2020). This model categorizes procrastination into two dimensions or different types of behaviors: (1) soldiering and (2) cyberslacking. Soldiering (offline procrastination) refers to avoiding work tasks for extended periods (e.g., over an hour daily). Examples of soldiering include gossiping, daydreaming, or engaging in more pleasurable activities than working, such as taking long coffee breaks. Consequently, soldiering arises as an ineffectual behavior. Cyberslacking (online procrastination) is a form of procrastination at work that has emerged with the widespread use of (mobile) technology in workplaces. Employees might appear to be engaged with their work and working on their computers but might instead be shopping online, browsing social media, gaming, or instant-messaging. Studies report massive costs for companies due to cyberslacking. The losses associated with cyberslacking include not only reduced performance but also diminished network security, slower network performance, and costs associated with removing viruses and spyware. The wide utilization of the Internet on company computers and personal mobile devices facilitates cyberslacking activities among employees and is becoming increasingly popular among employees (Metin et al., 2020; Yan & Zhang, 2022).

1.1.2. Authentic Leadership (AL) and Employees' Procrastination

Authentic Leadership (AL) can be defined as a leader's behavior pattern that achieves a performance beyond expectations, which is sustainable and maintained over time, as a consequence of the leaders' relationship with their collaborators in the organizations where they work (Avolio & Gardner, 2005; Luthans & Avolio, 2003). AL is composed of four dimensions (Walumbwa et al., 2008): (1) Self-awareness. This aspect of authentic leaders involves knowing their strengths, areas for improvement, values, emotions, and motivations, as well as contradictory aspects, biases, and defense mechanisms. However, most especially it involves being aware of how their own actions and behaviors have a decisive influence on coworkers and the context (Gardner et al., 2005; Ilies et al., 2005). (2) Balanced processing refers to the fact that authentic leaders objectively analyze data and facts, both external and related to themselves when making a decision. It assumes that these leaders do not distort their approaches and decisions for reasons of self-defense and/or self-enhancement, involving emotional and cognitive self-regulation skills (Ilies et al., 2005). (3) Moral perspective implies that when authentic leaders have to make a decision, the criteria they will use are their moral values. The behavior of authentic leaders is grounded on moral and ethical standards, even when faced with potential group, social, or organizational pressures; it produces ethical and transparent behaviors aimed at assisting the common group interests, which are sometimes in direct conflict with the leader's own

personal interests (Ilies et al., 2005). (4) Relational transparency involves the full sharing of information. It would include admitting mistakes when they are made, encouraging each person to express their opinion, and always telling the truth, even if it is hard. This dimension is also reflected through behaviors that encourage and reinforce group members to 'positively critique' and suggest ways to improve aspects of the work, fostering a culture of openness that, in turn, enhances learning (Avolio & Gardner, 2005; Gardner et al., 2005; Luthans & Avolio, 2003; Walumbwa et al., 2008). In addition to these four dimensions, authentic leaders also influence employees through five mechanisms, namely, positive modeling, personal and group identification, positive emotional contagion, support for self-determination, and positive social exchanges (Ilies et al., 2005).

Then, we propose that AL may directly reduce employees' procrastinatory behaviors for several reasons. First, authentic leaders could play a key role in preventing inappropriate work behaviors because they must lead by example. By being role models of ethical and disciplined work, they encourage team members to emulate their commitment, minimizing procrastinatory behavior. Moreover, authentic leaders are future-oriented (Gardner et al., 2005); that is, they are proactive and take a long-term view, so through positive modeling, they will develop this proactive attitude in their followers. Secondly, authentic leaders are clear and transparent in communicating goals and expectations, which helps team members understand the importance of their tasks. This reduces ambiguity, a known trigger for procrastination (Hoppe et al., 2018), by removing doubts about priorities and expected outcomes. Moreover, through relational transparency and positive interchanges with their collaborators, authentic leaders could help employees to become competent in the job by giving them constructive feedback and, thus, reducing procrastination due to task difficulty. Third, AL could negatively influence employees' procrastination behavior through employees' intrinsic motivation and engagement. That is, AL creates a need-supportive environment that could satisfy employees' basic psychological needs. According to Self-Determination Theory (SDT), when the need is satisfied, employees' intrinsic motivation and engagement will be increased and strengthened, encouraging them to enact more positive behaviors (Gagné & Deci, 2005) and cut down negative behaviors like procrastination. In fact, AL has been associated with employees' psychological needs, satisfaction, and engagement (Giallonardo et al., 2010; Hwang et al., 2022; Leroy et al., 2015; Schoofs et al., 2024). Empirical evidence suggests humble and inclusive leadership, which are features of AL (Avolio et al., 2004; Gardner et al., 2005; Ilies et al., 2005; Walumbwa et al., 2008), negatively related to employee procrastination (He et al., 2023; Liao et al., 2023; Lin, 2018). Based on these arguments, we hypothesize the following:

**Hypothesis 1 (H1).** *AL will be negatively associated with employees' procrastination at work:*

**H1a.** *AL will be negatively associated with employees' soldiering.*

**H1b.** *AL will be negatively associated with employees' cyberslacking.*

### 1.1.3. AL and Lack of Leader Support

Lack of leader support refers to employees' perception that their leader fails to provide adequate guidance, emotional support, or resources in the performance of their tasks. This can manifest ineffective communication, indifference to the needs of the team, or limited availability of the leader (Schmidt et al., 2018). Authentic leaders foster open and honest relationships with their team, creating an environment where workers feel valued and listened to. Specifically, authentic leaders tend to develop high-quality relationships based on the principles of social exchange rather than economic exchange through empowerment and support to the subordinates. The relationships that authentic leaders establish with their

collaborators are characterized by high levels of respect, assistance, and trust, reinforcing the perception of psychological safety (Ilies et al., 2005) and thus removing barriers associated with lack of guidance or support. Empirical studies show that AL is positively related to employees' psychological empowerment, psychological safety, and satisfaction with their leader (Hoch et al., 2018; Zhang et al., 2022). We therefore hypothesize the following:

**Hypothesis 2 (H2).** *AL will be negatively associated with a lack of leader support.*

1.1.4. AL and Lack of Workgroup Support

Workgroup support plays a crucial role in the work environment as it influences the perception of social connectedness and group cohesion. If employees perceive that their colleagues are not collaborative or supportive, this can increase their sense of emotional exhaustion, depersonalization, and anxiety while reducing their perception of psychological safety (Farfán et al., 2019; Rauf et al., 2024; Vieira et al., 2024). AL fosters an environment of psychological safety that not only improves the leader-follower relationship but also enhances collaboration among teammates (Zhang et al., 2022). In fact, Kahn (1990) stresses that such safe environments strengthen interpersonal relationships and reduce the perception of lack of support within the group. According to Ilies et al. (2005), when leaders are genuine and concerned about collective well-being, employees tend to imitate these behaviors, strengthening support networks within the team, thus reducing perceptions of lack of support. Authentic leaders, genuinely engaging with employees and their needs, foster a sense of belonging that strengthens group cohesion (Gardner et al., 2005). Empirical evidence links AL to greater workgroup identification (Edú-Valsania et al., 2016b) and cohesion in the workgroup (García-Guiu et al., 2015). Thus, we hypothesize the following:

**Hypothesis 3 (H3).** *AL will be negatively associated with a lack of workgroup support.*

1.1.5. Lack of Leader Support and Employees' Procrastination

Organizational Commitment Theory (Meyer & Allen, 1991) posits that perceived leader support increases employees' identification with the organization, thereby increasing their commitment and motivation. Conversely, the absence of such support detrimentally affects these factors, reducing employees' willingness to fulfill their responsibilities and increasing the likelihood of procrastination. Complementarily, the SDT (Deci & Ryan, 1985, 2014) asserts that employees must feel valued and supported by their leaders to maintain high levels of intrinsic motivation. When this support is lacking, employees are prone to demotivation. This will result in a reduced fulfillment of tasks and responsibilities.

From the perspective of the Job Demands-Resources Theory (Bakker & Demerouti, 2007), a lack of leader support represents an insufficiency of psychosocial resources, heightening the risk of workplace stress. Moreover, the lack of leader support generates uncertainty and demotivation (Bakker & Demerouti, 2013; Spielberger et al., 2003), factors closely linked to procrastination emotions (Bellini et al., 2022; Majeed et al., 2023; Sirois, 2023). Empirically, it has been shown that environments with limited resources and excessive demands can lead employees to engage in counterproductive behaviors, such as procrastination (Metin et al., 2016). In this context, the absence of a leader's support can be seen as an excessive demand and a significant stressor affecting employees' procrastination. Thus, we hypothesize the following:

**Hypothesis 4 (H4).** *Lack of leader support will be positively associated with employees' procrastination at work:*

**H4a.** *Lack of leader support will be positively associated with employees' soldiering.*

**H4b.** *Lack of leader support will be positively associated with employees' cyberslacking.*

1.1.6. Lack of Workgroup Support and Employees' Procrastination

Social relationships based on distrust and lack of reciprocity are predictors of stress and negative emotions at work (Jolly et al., 2021; Sirois, 2023). These negative emotions not only affect employees' personal experiences but are also positively associated with procrastination (Sirois, 2023). In a work environment lacking group support, employees may experience feelings of isolation and disengagement (Spielberger et al., 2003) and decrease their workgroup identification and sense of belonging to it (Tajfel & Turner, 1979). Consequently, employees may be less motivated to make an effort to achieve common goals.

On the other hand, a workgroup with strong supportive relationships among its members fosters a positive social learning process where members tend to model positive behaviors of their peers (Bandura, 1977, 1999). However, when this peer support is lacking, the process is broken. Additionally, the absence of peer support may also make it difficult to solve complex tasks. This significantly increases the likelihood that they will procrastinate, especially in ambiguous tasks that require greater effort (Hoppe et al., 2018). Complementarily, peer support acts as an immediate reinforcement and positive pressure, motivating employees to accomplish their tasks. Without the workgroup support employees would be less motivated as they lack social reinforcement. In fact, a lack of support and cohesion among coworkers is related to an increase in employee procrastination (Bellini et al., 2022). We therefore hypothesize the following:

**Hypothesis 5 (H5).** *Lack of workgroup support will be positively associated with employees' procrastination at work:*

**H5a.** *Lack of workgroup support will be positively associated with employees' soldiering.*

**H5b.** *Lack of workgroup support will be positively associated with employees' cyberslacking.*

1.1.7. Indirect Effects of AL on Employees' Procrastination

Employees' perceptions of a lack of leader support trigger negative emotions that, in turn, facilitate procrastination (Sirois, 2023). In contrast, AL builds high-quality relationships with subordinates based on respect, helpfulness, and empowerment (Avolio et al., 2004; Ilies et al., 2005), which are incompatible with perceptions of insufficient support. As a result, adopting an AL style is likely to reduce employees' perceptions of a lack of support, thereby indirectly mitigating procrastination behaviors. Based on this rationale, we propose that the effect of AL on employee procrastination is mediated by its ability to lower perceptions of a lack of leader support. Accordingly, we hypothesize the following:

**Hypothesis 6 (H6).** *Lack of leader support will mediate the relationship between AL and employees' procrastination:*

**H6a.** *Lack of leader support will mediate the relationship between AL and employees' soldiering.*

**H6b.** *Lack of leader support will mediate the relationship between AL and employees' cyberslacking.*

Lack of support within the workgroup becomes a stress factor for employees (Spielberger & Reheiser, 2020), which generates a feeling of disengagement, discomfort, and demotivation, thus increasing the likelihood of procrastination (Bellini et al., 2022; Sirois, 2023). AL increases the level of commitment and motivation among employees (Avolio et al., 2004; Ilies et al., 2005; Gardner et al., 2011) and is positively associated with the

cohesion of team members (García-Guiu et al., 2015), identification with the workgroup (Edú-Valsania et al., 2016b) and collaboration among professionals (Regan et al., 2016). Aspects that are in opposition to the lack of support from the workgroup. These studies suggest that AL has the potential to reduce the lack of support in the workgroup. When reducing this lack of support, since a lack of workgroup support would have positive effects on employee procrastination, these behaviors should also be indirectly reduced by AL. Thus, the effect of AL on employee procrastination is also indirect by taking the form of a reduction in the lack of support in the workgroup, so that when a potential trigger for procrastination is directly reduced, procrastination behavior is indirectly and negatively influenced by AL. Thus, we hypothesize the following:

**Hypothesis 7 (H7).** *Lack of workgroup support will mediate the relationship between AL and employees' procrastination:*

**H7a.** *Lack of workgroup support will mediate the relationship between AL and employees' soldiering.*

**H7b.** *Lack of workgroup support will mediate the relationship between AL and employees' cyberslacking.*

Figure 1 provides a graphical summary of the hypotheses described in the preceding sections, illustrating the expected theoretical relationships among the study variables.

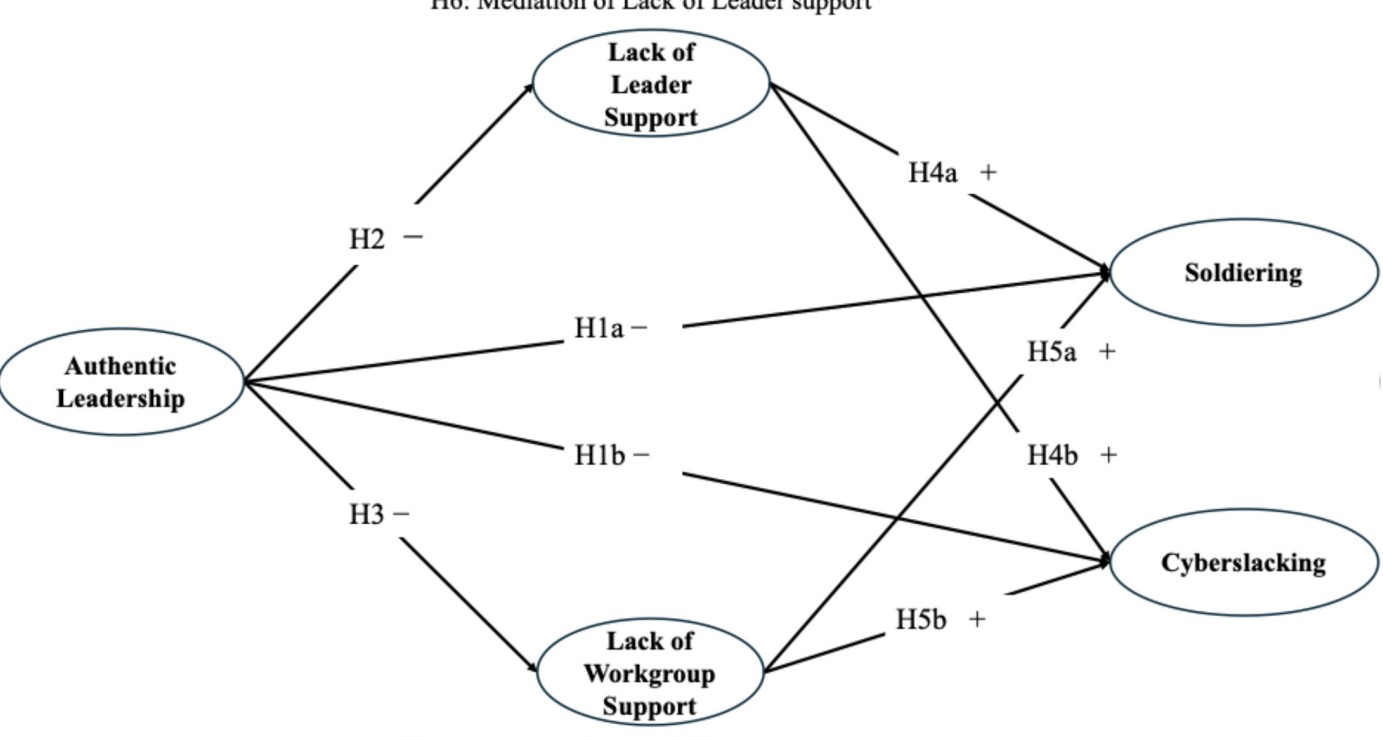

**Figure 1.** Theoretical model and expected relationships.

## 2. Materials and Methods

### 2.1. Participants and Procedure

For this cross-sectional, non-experimental, and correlational study, we used a convenience sampling method. While no strict inclusion or exclusion criteria were established, participants were required to be employed at the time of the study. The sample consisted of

738 employees (62.9% female) with an average age of 35.8 years (SD = 9.93; range = 18–62; Table 1). Most participants had a university education (57.2%). Participants were employed in Spanish organizations, both private (81.8%) and public (18.2%), primarily located in Madrid (33.2%), Andalusia (10.2%), and Castile and Leon (9.6%). The organizations represented various sectors, with the service (35%), health (12%), and education (11%) sectors being the most frequent. Among these organizations, 46.8% were large (over 250 employees), 27% were medium-sized (between 50 and 250 employees), 16.8% were small (from 10 to 49 employees), and 8.3% were micro-organizations (fewer than 10 employees).

**Table 1.** Sample characteristics.

| | | |
|---|---|---|
| Age | Mean (range) | 35.82 (18–62) |
| | SD | 9.93 |
| Gender | Male | 37.1% (270) |
| | Female | 62.9% (457) |
| Educational Level | Basic Education | 8.0% |
| | Secondary Education | 34.8% |
| | University | 57.2% |
| Type of Organization | Private | 81.8% |
| | Public | 18.2% |
| Sector of Activity | Services | 35.0% |
| | Health | 12.0% |
| | Education | 11.0% |
| | Industry and Technology | 10.5% |
| | Administration | 9.0% |
| | Commerce | 7.0% |
| | Hospitality | 7.0% |
| | Marketing | 6.0% |
| | Banking | 2.5% |
| Organizational Size | Micro (less than 10 employees) | 8.3% |
| | Small (10 to 49 employees) | 16.8% |
| | Medium (50 to 249 employees) | 27.0% |
| | Large (more than 250 employees) | 46.8% |
| Location | Com. Madrid | 33.20% (245) |
| | Andalusia | 10.30% (76) |
| | Castile and Leon | 9.89% (73) |
| | Canary Islands | 8.27% (61) |
| | Catalonia | 7.86% (58) |
| | Basque Country | 6.50% (48) |
| | Valencian Community | 5.69% (42) |
| | Castile-La Mancha | 4.88% (36) |
| | Com. Cantabria | 4.61% (34) |
| | Galizia | 4.07% (30) |
| | Navarra | 1.90% (14) |
| | Ceuta and Melilla | 1.36% (10) |
| | Balearic Islands | 1.22% (9) |
| | Extremadura | 0.41% (3) |
| | Aragon | 0.27% (2) |

Participants completed an online questionnaire in Spanish measuring the variables of the study. They were informed about the study's purpose, voluntary nature, data confidentiality, anonymity, and their right to withdraw at any time without penalty. These participants were recruited by MSc students in People Management, who administered the questionnaire to 10 employees each from different organizations. Although participants were part of workgroups, this study focuses on their individual perceptions, so the level of analysis is individual. To address potential common method bias, procedural remedies were

implemented, including emphasizing anonymity and designing a concise questionnaire that required approximately 15 min to complete.

*2.2. Instruments*

2.2.1. Authentic Leadership

We used the Spanish version (Moriano et al., 2011) of the Authentic Leadership Questionnaire (ALQ; Walumbwa et al., 2008), which comprises 16 items rated on a Likert-type scale, asking participants to indicate how frequently each statement describes their supervisor's leadership style, ranging from 0 (Not at all) to 6 (Always or almost always). Reliability was excellent (Cronbach's $\alpha$ = 0.96; composite reliability rho-c = 0.95).

2.2.2. Lack of Leader Support

Four items (e.g., "My leader lacks confidence in my job performance") were selected from the Spanish adaptation (Medina-Aguilar et al., 2007; Suárez-Tunanñaña, 2013) of the ILO-WHO work stress scale. Responses were rated on a Likert-type scale ranging from 0 (Never) to 6 (Always). Reliability was acceptable (Cronbach's $\alpha$ = 0.77; composite reliability rho-c = 0.86).

2.2.3. Lack of Workgroup Support

Three items (e.g., "My coworkers do not offer me enough protection from the unfair work demands of the bosses") were also selected from the Spanish adaptation (Medina-Aguilar et al., 2007; Suárez-Tunanñaña, 2013) of the ILO-WHO work stress scale. Responses ranged from 0 (Never) to 6 (Always). Reliability was adequate (Cronbach's $\alpha$ = 0.71; composite reliability rho-c = 0.84).

2.2.4. Procrastination at Work

We used the Spanish version (Guzmán Alvarado & Rosales Palacios, 2017) of the Procrastination at Work Scale (PAWS; Metin et al., 2016). PAWS contains 12 items rated on a Likert-type scale, with responses ranging from 0 (Not at all) to 6 (Always or almost always). The scale includes the two above-mentioned factors: (1) soldiering (8 items, e.g., "I delay some of my tasks just because I do not enjoy doing them"), and (2) cyberslacking (4 items, e.g., "I spend more than half an hour on social network sites—Facebook, Instagram, Twitter, etc.—at work per day for leisure purpose").

As this scale has not yet been validated in the Spanish population, we consider it interesting and necessary to test the bifactorial structure of the Procrastination at Work Scale (PAWS; Metin et al., 2016) (Appendix A). The Spanish adaptation of the Procrastination at Work Scale is composed of the two factors proposed by Metin et al. (2016, 2020): (1) the soldiering subscale with seven items (reduced from eight), and (2) the cyberslacking subscale (the same as the original). Reliability for both scales was satisfactory (soldiering: Cronbach's $\alpha$ = 0.84 and composite reliability rho-c = 0.88; and cyberslacking: $\alpha$ = 0.83 and rho-c = 0.89).

2.2.5. Sociodemographic and Control Variables

We included in the study several control variables. On an individual level, these included gender, age, and educational level. On an organizational level, these were organization type, size, and activity sector.

*2.3. Data Analysis*

For data analysis, we employed partial least squares structural equation modeling (PLS-SEM). This technique is particularly suitable for small, medium, and large sample sizes, does not require assumptions about data normality, and allows for the analysis of

complex models with simultaneous relationships between latent variables and multiple indicators (Henseler et al., 2015).

Analyses were performed using SmartPLS version 4.0 (Ringle et al., 2024). Statistical significance was assessed using the bootstrapping method, generating 5000 samples from a total of 738 cases. To determine statistical significance, a critical t-value of 1.96 was applied, corresponding to a 95% confidence level ($p < 0.05$). Study data are available at Supplementary Materials.

## 3. Results

First, to ensure the validity of our findings, we examined the potential common method bias (CMB). CMB is a widespread concern in social science research (Podsakoff et al., 2012) that must be controlled for. Additionally, we implemented measures to mitigate the influence of social desirability bias, such as ensuring the anonymity of the questionnaire. One of the most widely used techniques in social research to detect CMB is Harman's single-factor test. This procedure involves performing an exploratory factor analysis (EFA) using all the items from the questionnaire. CMB is considered problematic when the first extracted factor accounts for more than 50% of the total variance. In our study, the results indicated that the first extracted factor accounted for only 30.4% of the total variance. Therefore, we conclude that CMB did not significantly affect our results, and further analyses are appropriate to deepen the understanding of the relationships between the variables included in our study model.

### 3.1. Descriptives

The descriptive results (Table 2) revealed moderate levels of AL in the study sample and low levels of employee procrastination at work, including both soldiering and cyberslacking. Mean scores indicated slightly higher levels of soldiering compared to cyberslacking, though variability was greater for the latter. Correlational analyses showed negative relationships between AL and both forms of employee procrastination, as well as between AL and lack of leader and workgroup support. Conversely, employee procrastination was positively associated with both a lack of leader support and a lack of workgroup support. These results provide initial support to our hypotheses and justify further analyses.

**Table 2.** Descriptive statistics and correlations.

| | M | SD | AVE | 1 | 2 | 3 | 4 | 5 | 6 | 7 | 8 |
|---|---|---|---|---|---|---|---|---|---|---|---|---|
| 1. Age | 35.78 | 9.91 | - | | | | | | | | |
| 2. Education level | 4.10 | 1.30 | - | −0.15 ** | | | | | | | |
| 3. Organizational size | 3.13 | 0.98 | - | 0.08 | 0.12 | | | | | | |
| 4. Authentic leadership | 3.75 | 1.23 | 0.54 | −0.03 | 0.09 * | −0.01 | *0.73* | | | | |
| 5. Lack of leader support | 1.73 | 1.30 | 0.60 | 0.06 | −0.13 ** | 0.03 | −0.42 ** | *0.78* | | | |
| 6. Lack of workgroup support | 1.57 | 1.26 | 0.63 | 0.06 | −0.14 ** | −0.02 | −0.30 ** | 0.62 ** | *0.79* | | |
| 7. PR_Soldiering | 1.55 | 1.04 | 0.52 | −0.06 | 0.08 ** | −0.04 | −0.24 ** | 0.34 ** | 0.37 ** | *0.72* | |
| 8. PR_Cyberslacking | 1.39 | 1.21 | 0.67 | −0.06 | 0.10 ** | −0.03 | −0.17 ** | 0.18 ** | 0.17 ** | 0.50 ** | *0.82* |

Note. PR: procrastination. ** $p < 0.001$. * $p < 0.05$. AVE: average extracted variance. The square root of the AVE of each variable is indicated on the diagonal in bold italics.

### 3.2. Measurement Models: Convergent and Discriminant Validity

To analyze the results, we followed a two-step analysis (Hair et al., 2017). First, we examined the measurement models, and second, we analyzed the structural model (validation of hypotheses). Regarding the first step, all of the factor loadings were significant. Reliability (Cronbach's alpha and composite reliability) was satisfactory in all the scales, with values exceeding 0.70, as detailed in the Instruments Section for each construct. To assess convergent validity, the average extracted variance (AVE) was used. The AVE for

each construct was acceptable, as all values were higher than 0.50, thus indicating that each construct explains at least 50% of the variance of its indicators (Table 2). With regard to discriminant validity, we followed the criterion of Fornell and Larcker (1981); we checked that the square root of the AVE of each construct (which is indicated on the diagonal of Table 2, in bold italics) was higher than the correlations of that construct with the other study variables. Additionally, following recent recommendations in PLS-SEM analysis (Hair et al., 2019), we ensured the heterotrait–monotrait ratio of correlations (HTMT) was below 0.85 (although values below 0.90 represent adequate validity, we took a more conservative approach by considering a value of 0.85, which is the recommended value when constructs are conceptually different). The higher HTMT = 0.83 corresponded to a lack of workgroup support and a lack of leader support, which are two related constructs. The other ratios were low ($HTMT_{AL\text{-}cyberslacking}$ = 0.19; $HTMT_{AL\text{-}soldiering}$ = 0.27; $HTMT_{AL\text{-}lack\ of\ leader\ support}$ = 0.49; $HTMT_{AL\text{-}lack\ of\ workgroup\ support}$ = 0.37; $HTMT_{lack\ of\ leader\ support\text{-}cyberslacking}$ = 0.23; $HTMT_{lack\ of\ leader\ support\text{-}soldiering}$ = 0.43; $HTMT_{lack\ of\ workgroup\ support\text{-}cyberslacking}$ = 0.23; $HTMT_{lack\ of\ workgroup\ support\text{-}soldiering}$ = 0.48; $HTMT_{cyberslacking\text{-}soldiering}$ = 0.59).

*3.3. Validation of Hypotheses*

The direct effects of AL on soldiering and cyberslacking (Table 3) were statistically significant. For soldiering, the negative coefficient (H1a: β = −0.12, $p < 0.05$) indicates that higher levels of AL are associated with lower levels of soldiering. Similarly, for cyberslacking, the negative coefficient (H1b: β = −0.12, $p < 0.05$) suggests that as AL increases, cyberslacking decreases. These direct effects support the substantial impact of AL in reducing both forms of procrastination, providing full support for H1.

**Table 3.** Direct Effects of AL.

| | Estimate | Std. Dev. | t-Value | *p* | 95% Confidence Interval | |
|---|---|---|---|---|---|---|
| | | | | | Lower | Upper |
| AL → Soldiering | −0.12 | 0.04 | 2.79 | 0.005 | −0.19 | −0.03 |
| AL → Cyberslacking | −0.12 | 0.04 | 2.41 | 0.005 | −0.19 | −0.03 |

Note. AL: Authentic Leadership.

Standardized path coefficients (Table 4) offered further insight into the mediating and control variables. Regarding the relationships between AL and mediating variables, AL negatively affected both the lack of leader support (H2: β = −0.43, $p < 0.001$) and the lack of workgroup support (H3: β = −0.30, $p < 0.001$). Thus, higher AL scores were associated with lower perceptions of lack of support from both the leaders and workgroups, fully supporting H2 and H3. Regarding the mediators' relationships with employee procrastination, lack of leader support positively related to soldiering (H4a: β = 0.14, $p < 0.01$) but did not significantly predict cyberslacking (H4b: β = 0.10, $p = 0.065$). Lack of workgroup support positively related to soldiering (H5a: β = 0.27, $p < 0.001$), but not cyberslacking (H5b: β = 0.10, $p = 0.053$). These findings indicate that both mediators are strong predictors of soldiering but not cyberslacking, providing partial support to H5 and H6.

**Table 4.** Path coefficients.

| | Estimate | Std. Dev | t-Value | *p* | 95% Confidence Interval | |
|---|---|---|---|---|---|---|
| | | | | | Lower | Upper |
| AL → Lack of leader support | −0.42 | 0.04 | 11.13 | <0.001 | −0.49 | −0.34 |
| AL → Lack of workgroup support | −0.30 | 0.04 | 8.02 | <0.001 | −0.37 | −0.23 |
| Lack of leader support → Soldiering | 0.15 | 0.05 | 2.93 | <0.01 | 0.05 | 0.25 |
| Lack of workgroup support → Soldiering | 0.27 | 0.05 | 5.89 | <0.001 | 0.17 | 0.36 |
| Lack of leader support → Cyberslacking | 0.10 | 0.05 | 1.85 | 0.065 | −0.01 | 0.20 |
| Lack of workgroup support → Cyberslacking | 0.10 | 0.05 | 1.94 | 0.053 | −0.01 | 0.20 |
| Age → AL | −0.01 | 0.04 | 0.39 | 0.698 | −0.09 | 0.06 |
| Age → Cyberslacking | −0.04 | 0.04 | 1.27 | 0.206 | −0.11 | 0.02 |
| Age → Lack of leader support | 0.03 | 0.04 | 0.81 | 0.419 | −0.04 | 0.10 |
| Age → Lack of workgroup support | 0.04 | 0.04 | 0.89 | 0.373 | −0.04 | 0.11 |
| Age → Soldiering | −0.06 | 0.03 | 2.00 | 0.045 | −0.13 | 0.00 |
| Education Level → AL | 0.09 | 0.04 | 2.27 | 0.018 | 0.01 | 0.16 |
| Education Level → Cyberslacking | 0.13 | 0.04 | 3.48 | 0.001 | 0.05 | 0.21 |
| Education Level → Lack of leader support | −0.10 | 0.04 | 2.63 | 0.009 | −0.18 | −0.03 |
| Education Level → Lack of workgroup support | −0.11 | 0.04 | 2.70 | 0.007 | −0.19 | −0.03 |
| Education Level → Soldiering | 0.13 | 0.04 | 3.70 | 0.000 | 0.06 | 0.20 |
| Gender → AL | 0.07 | 0.08 | 0.95 | 0.344 | −0.07 | 0.23 |
| Gender → Cyberslacking | 0.02 | 0.07 | 0.31 | 0.754 | −0.13 | 0.16 |
| Gender → Lack of leader support | 0.09 | 0.07 | 1.40 | 0.160 | −0.04 | 0.22 |
| Gender → Lack of workgroup support | 0.09 | 0.07 | 1.32 | 0.188 | −0.05 | 0.23 |
| Gender → Soldiering | 0.08 | 0.07 | 1.22 | 0.222 | −0.05 | 0.22 |
| Organizational Size → AL | −0.02 | 0.04 | 0.46 | 0.647 | −0.09 | 0.05 |
| Organizational Size → Cyberslacking | −0.04 | 0.04 | 1.10 | 0.272 | −0.12 | 0.03 |
| Organizational Size → Lack of leader support | 0.03 | 0.03 | 1.00 | 0.315 | −0.03 | 0.09 |
| Organizational Size → Lack of workgroup support | 0.00 | 0.04 | 0.02 | 0.981 | −0.07 | 0.07 |
| Organizational Size → Soldiering | −0.06 | 0.03 | 1.75 | 0.081 | −0.13 | 0.01 |

Note. AL: Authentic Leadership.

As for socio-demographic and control variables, gender and organizational size showed no significant relationships with procrastination (both soldiering and cyberslacking). However, age had a slightly significant and negative impact on soldiering ($\beta = -0.06$, $p < 0.05$) but not cyberslacking. Results also show that employees with a higher education level have a greater tendency to procrastinate at work, both soldiering ($\beta = 0.13$, $p < 0.001$) and cyberslacking ($\beta = 0.13$, $p < 0.01$). Education level also positively influenced perceptions of AL ($\beta = 0.09$, $p < 0.05$) and negatively influenced lack of leader support ($\beta = -0.10$, $p < 0.01$) and lack of workgroup support ($\beta = -0.11$, $p < 0.01$).

Regarding the indirect individual pathways of AL on employee procrastination (see Table 5), we found that part of the negative relationship between AL and soldiering was mediated by reductions in lack of leader support (H6a: $\beta = -0.06$, $p < 0.05$) and lack of workgroup support (H7a: $\beta = -0.08$, $p < 0.001$). This indicates that improving AL reduces perceptions of lack of insufficient leader and workgroup support, which in turn decreases soldiering. However, lack of leader and workgroup support did not significantly mediate the relationship between AL and cyberslacking (H6b: $\beta = -0.04$, $p = 0.08$; H7b: $\beta = -0.03$, $p = 0.06$, respectively). When examining total indirect effects (see Table 6), we observed that the combined mediation effects of lack of leader and workgroup support significantly accounted for the relationship between AL and soldiering ($\beta = -0.14$, $p < 0.001$).

Although individual mediations were nonsignificant, the total indirect effect suggested that the simultaneous reduction in both forms of lack of support had a small but significant mediated relationship between AL and cyberslacking ($\beta = -0.07$, $p < 0.01$).

**Table 5.** Indirect effects of AL on employee procrastination.

| | Estimate | Std. Dev | t-Value | *p* | 95% Confidence Interval | |
|---|---|---|---|---|---|---|
| | | | | | Lower | Upper |
| AL →Lack of leader support → Soldiering | −0.06 | 0.02 | 2.61 | <0.05 | −0.11 | −0.02 |
| AL → Lack of workgroup support → Soldiering | −0.08 | 0.02 | 5.01 | <0.001 | −0.12 | −0.05 |
| AL → Lack of leader support →Cyberslacking | −0.04 | 0.02 | 1.76 | 0.08 | −0.09 | 0.00 |
| AL → Lack of workgroup support → Cyberslacking | −0.03 | 0.02 | 1.91 | 0.06 | −0.06 | 0.00 |

Note. AL: Authentic Leadership.

**Table 6.** Total indirect effects of AL on employee procrastination.

| | Estimate | Std. Dev. | t-Value | *p* | 95% Confidence Interval | |
|---|---|---|---|---|---|---|
| | | | | | Lower | Upper |
| AL → Soldiering | −0.14 | 0.03 | 5.84 | <0.001 | −0.19 | −0.10 |
| AL → Cyberslacking | −0.07 | 0.02 | 3.63 | <0.001 | −0.11 | −0.03 |

Note. AL: Authentic Leadership.

Finally, the total effects (direct + total indirect) of AL on employee procrastination (Table 7) were significant for both soldiering ($\beta = -0.26$, $p < 0.001$) and cyberslacking ($\beta = -0.19$, $p < 0.001$) and were stronger than the direct effects, confirming the indirect influence of AL on employee procrastination by reducing lack of support from leaders and workgroup and supporting H6 and H7.

**Table 7.** Total effects of AL on employee procrastination.

| | Estimate | Std. Dev | t-Value | *p* | 95% Confidence Interval | |
|---|---|---|---|---|---|---|
| | | | | | Lower | Upper |
| AL → Soldiering | −0.26 | 0.04 | 7.22 | <0.001 | −0.32 | −0.18 |
| AL → Cyberslacking | −0.19 | 0.04 | 5.22 | <0.001 | −0.26 | −0.11 |

Note. AL: Authentic Leadership.

Regarding the explanatory power of the model (see Table 8), the adjusted $R^2$ values show that variables of the study predict 19.4% of soldiering but only 6.5% of cyberslacking. On the other hand, study variables predict an 18.8% lack of leader support and a 10.5% lack of workgroup support. The control variables did not significantly predict LA. Finally, the $Q^2$ values for the endogenous construct were over 0 (Table 8); thus, the model has significant relevance (Hair et al., 2017).

**Table 8.** Explanatory power.

| | $Q^2$ **Predict** | $R^2$ **Adjusted** |
|---|---|---|
| Soldiering | 0.05 | 0.194 |
| Cyberslacking | 0.03 | 0.065 |
| Lack of leader support | 0.18 | 0.188 |
| Lack workgroup support | 0.06 | 0.105 |
| AL | | 0.05 ns |

Note. AL: Authentic Leadership. ns: non-significant.

Figure 2 summarizes these results.

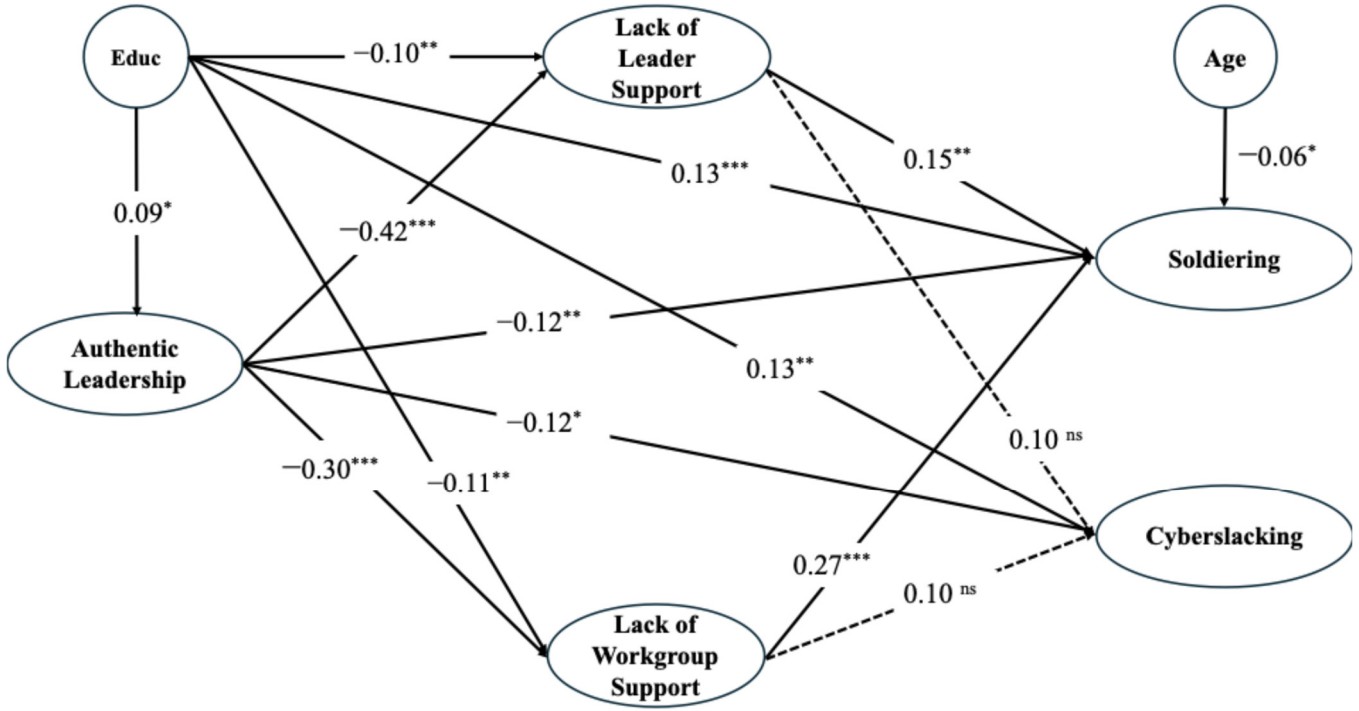

**Figure 2.** Summary of study results. Note. *** $p < 0.001$. ** $p < 0.01$. * $p < 0.05$. ns: non-significant. Educ: level of education.

## 4. Discussion

This study has been designed with a fourfold objective. Firstly, to explore the relationship between AL and employee procrastination behaviors at work. The results indicate that the more employees perceive their leader as authentic, the less they report engaging in procrastination behaviors. Secondly, we examined the relationship between AL and perceptions of lack of support from the leader and the workgroup. In line with this objective, AL is associated with greater perceived support from both the leader and the workgroup or, conversely, with lower levels of perceived lack of support. As for the third objective, i.e., to analyze the potential role of lack of supervisor and workgroup support in employees' procrastination at work, our findings show that lack of supervisor and workgroup support leads to higher levels of procrastination behaviors at work. Finally, the fourth objective was to investigate the possible mediating effects of lack of supervisor and workgroup support on the relationship between AL and employees' procrastination at work. Indeed, our results confirm that lack of support mediates the relationship between AL and employees' procrastination behaviors.

Our findings support the hypotheses formulated, showing both direct and indirect negative effects of AL on procrastination. Specifically, AL was negatively associated with

employees' procrastination at work (H1), including both employees' soldiering (H1a) and cyberslacking (H1b). Employees who perceived their leaders as more authentic were less likely to procrastinate at work, with a stronger negative effect on soldiering compared to cyberslacking. These findings align with previous studies that have linked effective leadership to reduced employee procrastination (He et al., 2023; Liao et al., 2023; Lin, 2018). Moreover, our study has shown that AL has a greater negative effect on the offline procrastination behaviors of employees (soldiering) than on their online procrastination behaviors (cyberslacking). This result could be explained because, nowadays, Internet use at work is growing, and it is increasingly common to use mobile technologies at work, which, in turn, significantly increases the likelihood of cyberslacking (Tandon et al., 2022; Venkatesh et al., 2023; Vitak et al., 2011). So, this professional usage can modulate or inhibit the effect of AL.

AL was also negatively associated with a lack of leader support (H2) and a lack of workgroup support (H3). The results further revealed that AL was negatively related to perceptions of lack of support from leaders and workgroups, such that the more leaders are perceived as authentic, the lower the employees' perception of lack of support from both their leader and coworkers. Notably, AL had a stronger effect on reducing the perceived lack of leader support than on reducing the lack of workgroup support. This difference could be explained because AL is potentially incompatible with the leader's lack of support for employees. Such leadership is characterized by building quality relationships with employees (Avolio et al., 2004; Ilies et al., 2005). However, although authentic leaders also foster a cohesive and supportive environment among the different members of the work unit, they are not fully capable of determining the type of interactions that members of the workgroup establish with each other since they are also influenced by aspects of an individual nature and personality that are completely unrelated to leadership and, which in turn, could influence the amount of support that group members provide to each other. The leader is more able to influence the amount of support they offer to coworkers than the amount of support coworkers offer to each other. Although previous studies have analyzed the relationship between AL and other variables related to leader support, such as employees' psychological empowerment, psychological safety, and satisfaction with their leader (Hoch et al., 2018; Zhang et al., 2022), or workgroup identification (Edú-Valsania et al., 2016b) and cohesion in the workgroup (García-Guiu et al., 2015), the present study goes a step further by specifically analyzing the relationship between AL and the perception of lack of support.

In this study, we also wanted to examine the effect that two specific risk factors might have on employees' procrastination behaviors —lack of support from the leader and lack of support from colleagues in the workgroup. Generally speaking, the results have shown that these two factors can indeed facilitate employees' procrastination at work. Specifically, however, it can be seen that these two factors only have a positive influence on offline procrastination or soldiering behaviors (H4a and H5a), and the effect of the lack of support from the group is greater than that of the leader. However, for online procrastination behaviors —cyberslacking—, lack of supervisor support and lack of peer support as risk factors, taken individually, do not have a statistically significant effect (H4b and H5b). It is possible that other factors such as task complexity, exhaustion, ease of access to the Internet (Tandon et al., 2022; Venkatesh et al., 2023; Vitak et al., 2011), and personal characteristics facilitate this type of online procrastination behavior (Koay & Poon, 2023). Hypotheses 6 and 7 examined the mediating role of lack of support from the leader and the workgroup, respectively, in the relationship between AL and procrastination (i.e., soldiering and cyberslacking). While lack of leader support and lack of workgroup support significantly mediated the relationship between AL and soldiering, these mediators individually did

not significantly influence the relationship between AL and cyberslacking. The absence of significant mediation effects for cyberslacking suggests the presence of other unexamined mediators. Future research should explore additional mediators and moderators that may influence cyberslacking. However, although individual mediation effects were nonsignificant, the total indirect effect suggested that the simultaneous reduction in both forms of lack of support had a small but significant mediated relationship between AL and cyberslacking. Thus, the total effects (direct + total indirect) of AL on employee procrastination were significant for both soldiering and cyberslacking and were stronger than the direct effects alone. These findings confirm the indirect influence of AL on employee procrastination by reducing the lack of support from leaders and workgroups.

This study has also shown that employees with a higher education level have a greater tendency to procrastinate at work (both soldiering and cyberslacking), consistent with the results of previous studies. Garrett and Danziger (2008) found that for higher-educated employees, higher levels of cyberslacking were associated. Education level also positively influenced perceptions of AL and negatively influenced lack of leader support, suggesting that employees with high education view their leaders as more authentic and supportive. Finally, our study also showed that the older the age of the participants, the lower their tendency to procrastinate offline. However, this variable has not been shown to be significant in predicting online procrastination or cyberslacking.

### 4.1. Limitations and Future Research

This study has several limitations. Its correlational and cross-sectional design prevents establishing causality, so experimental and longitudinal studies are needed to confirm the cause-effect relations between AL and employees' procrastination. As the scores of employees' procrastination consisted of the participants' self-reports of their own performance, the responses may be underestimated due to social desirability bias. Future studies could use other sources, for example, leaders' perceptions, to obtain these data. Additionally, the type of tasks and employees' personalities were not taken into account, and they could influence and modulate the effects of leadership on the employees' behavior. Lastly, we did not carry out stratification by activity sector of the participants, and this affects the external validity of the results.

The results of the present study and its limitations can guide future research. First, the relationship between other positive leadership styles (e.g., servant, ethical, transformational, secure base) and procrastination could be analyzed. Second, we recommend using mixed methods in the collection of data on procrastination to capture contextual nuances in procrastination. We also recommend that personality variables be considered in the analysis of individual procrastination. Third, it is important to analyze other possible mediating and moderating mechanisms in the relationship between leadership and procrastination. Leadership affects important group variables such as work climate or group cohesion, which could, in turn, play a mediating role. Furthermore, the present study has focused on individual analysis, although a specific study with workgroups could delve deeper into group variables, considering different levels of analysis to enrich the research.

### 4.2. Theoretical and Practical Implications

Despite the above-mentioned limitations, this study advances the understanding of AL's impact by showing its negative association with employee procrastination at work and lack of support in work environments. Thus, this study contributes to analyzing these relationships that have not yet been explored in the previous literature. On the other hand, this paper emphasizes the importance of considering procrastination not as a unidimensional measure but considering its two components (i.e., soldiering and

cyberslacking). Also, having a procrastination scale validated in Spanish and in the context of Spain with adequate psychometric properties may encourage future research in this context, whether in relation to AL or other leadership styles.

This study also has practical implications that organizations should take into account in order to prevent their employees' procrastination. Organizations should prioritize AL training to enhance leader authenticity and supportiveness, which can mitigate procrastination. Furthermore, fostering robust support networks within workgroups should be a strategic focus for managers to reduce procrastination behaviors and enhance workplace productivity.

### 4.3. Conclusions

This study highlights two significant risk factors for procrastination at work: lack of leader support and lack of workgroup support. The greater the perceived lack of support from both the workgroup and leader, the greater the tendency for employees to procrastinate at work, especially offline. AL has the potential to mitigate workplace procrastination both directly and also indirectly by reducing perceptions of insufficient support. Specifically, AL has a greater capacity to inhibit offline procrastination behaviors than online procrastination. Likewise, AL has the capacity to reduce the lack of support at work, but it has a greater capacity to reduce the lack of support from the leader than from the workgroup.

Although employees with a higher level of education feel less lack of support at work and perceive their leaders as more authentic, they also have a greater tendency to procrastinate. Organizations should focus on training their managers and leaders in AL to promote authenticity and supportiveness while fostering strong support networks within workgroups to enhance productivity and reduce procrastination behaviors.

**Supplementary Materials:** The data supporting the conclusions of this article can be downloaded at https://www.mdpi.com/article/10.3390/psycholint7010021/s1.

**Author Contributions:** Conceptualization, S.E.-V., A.L. and J.A.M.; methodology, S.E.-V.; formal analysis, S.E.-V.; investigation, S.E.-V.; resources, S.E.-V.; data curation, S.E.-V.; writing—original draft preparation, S.E.-V. and A.L.; writing—review and editing, S.E.-V., A.L. and J.A.M.; supervision, J.A.M. All authors have read and agreed to the published version of the manuscript.

**Funding:** This research received no external funding.

**Institutional Review Board Statement:** We conducted non-interventional research based on self-reported questionnaires without any direct manipulation of the participants. Ethical review and approval were waived for this study, because the research does not affect the fundamental rights (life, physical/psychic integrity, health, freedom/autonomy in any of its manifestations, personal dignity, etc.) of the subjects involved (on whom the research is based), and no personally identifiable data are used, according to the Research Ethics Committee of the Spanish National Distance University (UNED).

**Informed Consent Statement:** Due to the anonymity and confidentiality of the participants, only those who accepted informed consent proceeded to fill out the study questionnaire. Written informed consent from the participants was not required to participate in this study in accordance with the national legislation and the institutional requirements.

**Data Availability Statement:** The raw data supporting the conclusions of this article are contained within the article and Supplementary Materials.

**Acknowledgments:** The authors would like to thank all the participants for their collaboration in this study. During the preparation of this manuscript, the authors used ChatGPT in order to correct possible misspellings and errors or improve the wording. After using this tool/service, the

authors reviewed and edited the content as needed and take full responsibility for the content of the publication. All authors have read and consented to the acknowledgement.

**Conflicts of Interest:** The authors declare no conflicts of interest.

## Abbreviations

The following abbreviations are used in this manuscript:

AL    Authentic Leadership
PR    Procrastination

## Appendix A. Validation of the Procrastination at Work Scale

As this scale has not yet been validated in the Spanish population, we consider it interesting and necessary to test the bifactorial structure of the Procrastination at Work Scale (PAWS; Metin et al., 2016) in order to achieve greater consistency when analyzing the data and explaining the results. For this, we conducted a confirmatory factor analysis (CFA) using JASP software version 0.18.3.0 (JASP Team, 2024), an increasingly popular tool among researchers due to its functionality and open-access model (Garbutt et al., 2024; Kasprzak & Mudło-Głagolska, 2022; Molero Jurado et al., 2021; Xie et al., 2024). First, we tested a single-factor model, including all scale items. The results of the fit indices for this model were not satisfactory (Table A1). We then tested the two-factor structure proposed by Metin et al. (2016, 2020) for the subscales. While this model improved the fit indices, some indices—particularly RMSEA—remained below acceptable thresholds, which should be under 0.08 and over 0.05 (Steiger, 1990) (Table A1). Upon reviewing the factor loadings, we decided to remove item PAWS_7 ("I take long coffee breaks") from the soldiering subscale. Once this item was removed, we reran the CFA, which yielded satisfactory fit indices. Therefore, the Spanish adaptation of the Procrastination at Work Scale is composed of the two factors proposed by Metin et al. (2016, 2020): (1) the soldiering subscale with seven items (reduced from eight) and (2) the cyberslacking subscale, which remained unchanged. Reliability for both scales was satisfactory (soldiering: Cronbach's $\alpha$ = 0.84 and composite reliability rho-c = 0.88; and cyberslacking: $\alpha$ = 0.83 and rho-c = 0.89). All analyses reported in this article accounted for this modification.

**Table A1.** CFA: Procrastination at Work.

| Model | $\chi^2$ | df | RMSEA | IFI | TLI | CFI | GFI |
|---|---|---|---|---|---|---|---|
| 2 Factors without item PAWS7 | 237 | 43 | 0.078 | 0.94 | 0.92 | 0.94 | 0.94 |
| 2 Factors | 339 | 53 | 0.086 | 0.92 | 0.90 | 0.92 | 0.93 |
| 1 Factor | 908 | 64 | 0.147 | 0.77 | 0.71 | 0.76 | 0.78 |

Note. $\chi^2$: chi-square; df: degrees of freedom; RMSEA: root mean square error of approximation; IFI: incremental fit index; TLI: Tucker–Lewis Index; CFI: comparative fit index; GFI; goodness-of-fit index.

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
