# Peer review of "Interpersonal Dynamics of Authentic Leadership: Effects on Support Perception and Workplace Procrastination"

_2813-9844, doi:10.3390/psycholint7010021_

Round 1

Reviewer 1 Report

Dear Authors,

Thank you very much for allowing me to review your article. The topic is interesting. However, the article to be published needs important changes. I hope my suggestions will be useful to you.

Abstract

Revise taking into account all suggestions.

1. Introduction

Suggestions:

1.   It is important that the introduction specifies the value of your research. what does your study add to previous research? what specific knowledge gap does your research aim to fill? why is your research important? has the relationship between authentic leadership and procrastination been analyzed before? which authors have done so?

2.   This reference should be updated (Spielberger et al., 2003).

3.   It is a priority to include a separate section that contextualizes the research.

1.1. Theoretical Background and Hypotheses

1.1.1. Procrastination at Work

Suggestions:

1.   The section is good. However, they should improve their explanation by including other authors and other perspectives. Actually, they only take into account Metin el al. (2016,2020).

1.1.2. Authentic Leadership (AL) and Employees' Procrastination

Hypotheses H1 (H1a and H1b) are well justified. I have no suggestions.

1.1.3. AL and Lack of Leader Support

Hypothesis H2 is well justified. I have no suggestions.

1.1.4. AL and Lack of Workgroup Support

Hypothesis H3 is well justified. I have no suggestions.

1.1.5. Lack of Leader Support and Employees' Procrastination

Suggestions:

1.   Hypotheses H4a and H4b need much more specific justification. Introduce new authors.

1.1.6.     Lack of Workgroup Support and Employees' Procrastination.

Suggestions:

1.   Hypotheses H5a and H5b need much more specific justification. Introduce new authors.

1.1.7. Indirect Effects of AL on Employees' Procrastination

Suggestions:

1.   Hypotheses H6 (H6a and H6b) and hypotheses H7 (H7a and H7b) need much more justification. In fact, indirect effects in any research are very important. They must adequately justify how or why the effect occurs and through what mechanisms.  

2.   Figure 1 should contain all the hypothesis not just the direction, positive or negative. Figure 1 should be much more professional. In addition, Figure 1 should be adequately explained. I suggest that you construct a separate section that can be called a theoretical model or study model.

2. Materials and Methods

Participants and Procedure

Suggestions:

1. Did you obtain informed consent from the participants? voluntary withdrawal? data confidentiality?

2. What type of sampling did you use?

3. Did you establish inclusion and exclusion criteria?

4. Did the study go through an ethics committee?

5. How did you address common method bias? Justify your answer much better. They include some data in the results, but it is totally insufficient.

6.   Explain whether your study is cross-sectional, non-experimental and correlational. Justify your choice.

2.2. Instruments

Suggestions:

1. All instruments should include AVE values.

2.   I understand that the questionnaire was entirely in Spanish, was it not?

3. Why did you not use control variables? This could be a limitation of the study. If they were used, they should vary the study model and include them in the instruments section.

2.3. Validation of the Procrastination at Work Scale

Suggestions:

1. I do not understand why you only did a confirmatory factor analysis of a single scale. Why did you not do CFA of the whole model? Justify your answer very well.

2. Why didn't you include convergent and discriminant validity of all the scales? Justify your answer very well.

3.   Include CFA as an annex.

2.4. Data Analysis

Suggestions:

1.      This section should include all analyses performed. For example, correlations between variables.

3. Results

Suggestions:

1.   In Table 3, instead of 1, include discriminant validity.

2.   They should include a section on validation of hypotheses.

3. I have not seen that they justify their hypotheses H4a and H4b; H5a and H5b; H6a and H6b; H7a and H7b.

4. Discussion

Suggestions:

1. Your research has 4 objectives and 7 main hypotheses. The discussion of results must include a justification of each of your objectives and each of your hypotheses based on previous studies. Do your results cover all 4 objectives? What do your hypotheses contribute? The discussion of results is one of the most important sections of a research study.

Limitations and Future Research

Suggestions:

1.   Limitations and future research are totally insufficient.

4.2. Contributions and Practical Implications

Suggestions:

1.   This section should be divided into theoretical implications and practical implications. It is a priority to include solid practical implications.

2.   The authors should include a section on conclusions.

Review specific comments made to the authors.

Author Response

Response to Reviewer 1 Comments

Major comment: Dear Authors, Thank you very much for allowing me to review your article. The topic is interesting. However, the article to be published needs important changes. I hope my suggestions will be useful to you. 

Thank you very much for your thoughtful review and constructive suggestions regarding our manuscript. We appreciate your recognition of the topic's interest and have diligently addressed all the important changes you suggested.

We have revised the manuscript accordingly and believe that these modifications have significantly enhanced the clarity and depth of our study. Below, you will find a detailed response to each of your suggestions, outlining how we have addressed them one by one in the revised manuscript. We are confident that these adjustments meet the high standards of the journal and effectively address your concerns.

Abstract. Revise taking into account all suggestions

We have carefully updated the abstract to reflect all the suggestions provided. The revised abstract now clearly emphasizes the role of Authentic Leadership (AL) as the primary variable, highlights the mediating effects of lack of supervisor and workgroup support, and aligns with the updated introduction and objectives of the study. Additionally, we have incorporated information about the methodology, specifying that data analysis was conducted using Partial Least Squares Structural Equation Modeling (PLS-SEM) to provide greater clarity and precision -following the comments noted by Reviewer 2, we have reanalyzed the data using another software-.

We hope the revised abstract now meets the expectations and effectively summarizes the key contributions of our study. 

Comments 1: Introduction: Suggestions: 

1.   It is important that the introduction specifies the value of your research. what does your study add to previous research? what specific knowledge gap does your research aim to fill? why is your research important? To the best of our knowledge the relationship between authentic leadership and procrastination has not yet been analyzed?

2.   This reference should be updated (Spielberger et al., 2003).

3.   It is a priority to include a separate section that contextualizes the research.

Response 1: Thank you very much for your valuable comments and suggestions regarding the introduction. We have carefully addressed all your points and made the following revisions, which are highlighted in red in the manuscript:

  1. The introduction has been revised to explicitly specify the value of our research, including the unique contributions of the study, the specific knowledge gap it addresses, and the importance of the research. We have clarified that the relationship between Authentic Leadership (AL) and procrastination behaviors has been unexplored in prior studies and we have framed our contribution more precisely by focusing on the mediating role of perceived support from supervisors and coworkers.
  2. The reference "Spielberger et al., 2003" has been updated to reflect the most recent and relevant literature.
  3. A separate section has been added to contextualize the research, providing a clearer framework and background for the study.

Comments 2: 1.1.1. Procrastination at Work. Suggestions: The section is good. However, they should improve their explanation by including other authors and other perspectives. Actually, they only take into account Metin el al. (2016,2020).

Response 2: We have carefully revised this section to include additional perspectives and references from other authors, expanding beyond Metin et al. (2016, 2020). These new contributions provide a broader and more comprehensive understanding of workplace procrastination, incorporating diverse views and findings from recent and relevant literature.

Comments 3: 1.1.5. Lack of Leader Support and Employees' Procrastination. Suggestions: Hypotheses H4a and H4b need much more specific justification. Introduce new authors.

Response 3: We have revised this section to improve the theoretical foundation and have introduced additional references to support these hypotheses more comprehensively. We believe these enhancements provide a stronger basis for the proposed hypotheses.

Comments 4: 1.1.6. Lack of Workgroup Support and Employees' Procrastination. Suggestions: Hypotheses H5a and H5b need much more specific justification. Introduce new authors.

Response 4: We have revised the section by providing a more detailed explanation to strengthen the justification for hypotheses H5a and H5b. The updates have been highlighted in red in the manuscript for your review.

Comments 5: 1.1.7. Indirect Effects of AL on Employees' Procrastination. Suggestions: Hypotheses H6 (H6a and H6b) and hypotheses H7 (H7a and H7b) need much more justification. In fact, indirect effects in any research are very important. They must adequately justify how or why the effect occurs and through what mechanisms Figure 1 should contain all the hypothesis not just the direction, positive or negative. Figure 1 should be much more professional. In addition, Figure 1 should be adequately explained. I suggest that you construct a separate section that can be called a theoretical model or study model.

Response 5: Thank you for your detailed comments regarding the section 1.1.7. Indirect Effects of AL on Employees' Procrastination. We have carefully addressed your suggestions as follows:

  1. We have expanded the justification for hypotheses H6 (H6a and H6b) and H7 (H7a and H7b), providing a clearer explanation of the mechanisms and theoretical reasoning underlying the indirect effects.
  2. Figure 1 has been revised to include all the hypotheses, providing a complete graphical representation of the study’s theoretical model.
  3. While we considered creating a separate section for the theoretical model, we determined that the updated figure and its accompanying explanation within the existing structure sufficiently address this need. The revised figure now effectively summarizes the hypotheses and theoretical relationships in a professional and comprehensive manner.

All changes are highlighted in red in the manuscript for your review. We hope these revisions meet your expectations and demonstrate our commitment to improving the clarity and rigor of the study.

Comments 6: Methods: Participants and Procedure

Suggestions:

1. Did you obtain informed consent from the participants? voluntary withdrawal? data confidentiality?

2. What type of sampling did you use?
3. Did you establish inclusion and exclusion criteria?

4. Did the study go through an ethics committee?

5. How did you address common method bias? Justify your answer much better. They include some data in the results, but it is totally insufficient. 

6.   Explain whether your study is cross-sectional, non-experimental and correlational. Justify your choice. 

Response 6: Thank you for your detailed feedback and questions regarding the Methods: Participants and Procedure section. Your comments have been instrumental in guiding us to improve this section. We have addressed each of your points as follows:

  1. We have clarified that informed consent was implicit in the online questionnaire. Participants indicated their acceptance by proceeding to the next screen, where the questionnaire began. This process ensured participants were informed of their rights, including voluntary withdrawal and data confidentiality, before participation.
  2. The type of sampling used has been explicitly described.
  3. We have detailed the inclusion and exclusion criteria applied in the study.
  4. Regarding ethical approval, we have included the following statement in the manuscript:

“Institutional Review Board Statement: We conducted non-interventional research based on self-reported questionnaires, without any direct manipulation of the participants. Ethical review and approval were waived for this study, as it does not affect the fundamental rights (life, physical/psychic integrity, health, freedom/autonomy in any of its manifestations, personal dignity, etc.) of the subjects involved, and no personally identifiable data is used. This is in accordance with the guidelines of the Research Ethics Committee at [University].”

  1. The issue of common method bias has been addressed more comprehensively, with additional justification provided in the revised text.
  2. We have explained that the study is cross-sectional, non-experimental, and correlational, and have included a justification for this methodological choice.

All changes have been highlighted in red in the manuscript for your review. We believe these revisions have significantly improved the clarity and rigor of this section.

Comments 7: 2.2. Instruments Suggestions:
1. All instruments should include AVE values.
2.   I understand that the questionnaire was entirely in Spanish, was it not?
3. Why did you not use control variables? This could be a limitation of the study. If they were used, they should vary the study model and include them in the instruments section. 

Response 7: Thank you for your detailed feedback on the 2.2. Instruments section. We have addressed your comments as follows:

  1. We have included the AVE values as suggested. In Table 3, the AVE is indicated on the diagonal, in bold italics.
  2. We have clarified in the Procedure section that the questionnaire was administered entirely in Spanish.
  3. Regarding control variables, these were already included in the manuscript under 2.2.5. Sociodemographic and Control Variables. Specifically, we considered several control variables: at the individual level, gender, age, and educational level; and at the organizational level, organization type, size, and activity sector. Additionally, these variables are detailed in Table 1. Sample characteristics, which was also included in the original manuscript and references these variables.

Comments 8: Validation of the Procrastination at Work Scale. Suggestions:

1. I do not understand why you only did a confirmatory factor analysis of a single scale. Why did you not do CFA of the whole model?

2. Why didn't you include convergent and discriminant validity of all the scales?

3.   Include CFA as an annex.

Response 8: Thank you for your comments on the Validation of the Procrastination at Work Scale section. Regarding your first suggestion, we would like to clarify that, as the scale has not yet been validated in the Spanish population, we considered it both interesting and necessary to test the bifactor structure of workplace procrastination. This approach was taken to ensure greater consistency when analyzing the data and explaining the results. This validation process is now detailed in Annex 1.

A new section entitled “Measurement models: convergent and discriminant validity” has been included. In this new section we explain convergent and discriminant validity of the study variables.

Comments 9: 2.4. Data Analysis. Suggestions: 1.      This section should include all analyses performed. For example, correlations between variables.

Response 9: We have revised this section to include a comprehensive overview of all analyses performed in our study. Specifically, we have updated Table 2. Descriptive Statistics and Correlations in the results section, which includes the correlations between variables. 

Comments 10: 3. Results. Suggestions:

1.   In Table 3, instead of 1, include discriminant validity.

2.   They should include a section on validation of hypotheses.

3. I have not seen that they justify their hypotheses H4a and H4b; H5a and H5b; H6a and H6b; H7a and H7b.

Response 10: We have revised this section to include a new analysis using SmartPLS software. A specific section on “Validation of hypotheses” has been included and we have tried our best to further clarify the results in relation to each of the hypotheses.

Comments 11: 4. Discussion. Suggestions:
1. Your research has 4 objectives and 7 main hypotheses. The discussion of results must include a justification of each of your objectives and each of your hypotheses based on previous studies. Do your results cover all 4 objectives? What do your hypotheses contribute? The discussion of results is one of the most important sections of a research study.

Response 11: We have revised this section in line with these comments. We hope you will agree that this section has improved substantially.

Comments 12: Limitations and Future Research. Suggestions: Limitations and future research are totally insufficient

Response 12: We have tried to expand this section in more detail.

Comments 13: 4.2. Contributions and Practical Implications. Suggestions: This section should be divided into theoretical implications and practical implications. It is a priority to include solid practical implications.

Response 13: In response to your feedback, we have revised this section and now distinctly separate the discussions into two specific paragraphs: one for theoretical implications and another for practical implications. This reorganization ensures that each type of implication is clearly outlined and discussed comprehensively.

We appreciate your guidance in emphasizing the importance of solid practical implications, and we have made it a priority to enhance this aspect of our manuscript.

We trust that these revisions meet your expectations and significantly strengthen the manuscript.

Comments 14: The authors should include a section on conclusions.

Response 14: In line with this appropriate suggestion, a final section on Conclusions has been added.

During the manuscript review process, the following new references have been incorporated (due to the use of a reference manager, they are not highlighted as new text in the manuscript):

Bakker, A. B. & Demerouti, E. (2007). The Job Demands‐Resources model: State of the art. Journal of Managerial Psychology, 22(3), 309–328. https://doi.org/10.1108/02683940710733115

Bandura, A. (1977). Social Learning Theory. Prentice-Hall.

Bandura, A. (1999). Social Cognitive Theory: An agentic perspective. Asian Journal of Social Psychology, 2(1), 21–41. https://doi.org/10.1111/1467-839X.00024

Deci, E. L. & Ryan, R. M. (1985). Toward an organismic integration theory. In Intrinsic motivation and self-determination in human behavior (pp. 113–148). Springer US. https://doi.org/10.1007/978-1-4899-2271-7_5

Deci, E. L. & Ryan, R. M. (2014). Autonomy and need satisfaction in close relationships: Relationships motivation theory. In N. Weinstein (Ed.), Human motivation and interpersonal relationships: Theory, research, and applications (pp. 53–73). Springer.

Hair, J. F., Hult, G. T. M., Ringle, Christian. M. & Sarstedt, M. (2017). A primer on Partial Least Squares Structural Equation Modeling (PLS-SEM) (2nd ed.). SAGE.

Henseler, J., Ringle, C. M. & Sarstedt, M. (2015). A new criterion for assessing discriminant validity in variance-based structural equation modeling. Journal of the Academy of Marketing Science, 43(1), 115–135. https://doi.org/10.1007/s11747-014-0403-8

Meyer, J. P. & Allen, N. J. (1991). A three-component conceptualization of organizational commitment. Human Resource Management Review, 1(1), 61–89. https://doi.org/10.1016/1053-4822(91)90011-Z

Prem, R., Scheel, T. E., Weigelt, O., Hoffmann, K. & Korunka, C. (2018). Procrastination in daily working life: A diary study on within-person processes that link work characteristics to workplace procrastination. Frontiers in Psychology, 9. https://doi.org/10.3389/fpsyg.2018.01087

Regan, S., Laschinger, H. K. S. & Wong, C. A. (2016). The influence of empowerment, authentic leadership, and professional practice environments on nurses’ perceived interprofessional collaboration. Journal of Nursing Management, 24(1), E54–E61. https://doi.org/10.1111/jonm.12288

Ringle, C. M., Wende, S. & Becker, J.-M. (2024). SmartPLS 4. SmartPLS GmbH. http://www.smartpls.com

Spielberger, C. D. & Reheiser, E. C. (2020). Measuring occupational stress: The job stress survey. In R. Crandall (Ed.), Occupational stress (pp. 51–69). CRC Press.

Steiger, J. H. (1990). Structural model evaluation and modification: An interval estimation approach. Multivariate Behavioral Research, 25(2), 173–180. https://doi.org/10.1207/s15327906mbr2502_4

Steinert, C., Heim, N. & Leichsenring, F. (2021). Procrastination, perfectionism, and other work-related mental problems: Prevalence, types, assessment, and treatment—A scoping review. Frontiers in Psychiatry, 12. https://doi.org/10.3389/fpsyt.2021.736776

Tajfel, H. & Turner, J. C. (1979). An integrative theory of intergroup conflict. In W. G. Austin & S. Worchel (Eds.), The social psychology of intergroup relations (pp. 37–47).

Yan, B. & Zhang, X. (2022). What research has been conducted on procrastination? Evidence from a systematical bibliometric analysis. Frontiers in Psychology, 13. https://doi.org/10.3389/fpsyg.2022.809044

Reviewer 2 Report

Read the following text:

Thank you for the opportunity to review the article "Interpersonal Dynamics of Authentic Leadership: Effects on Support Perception and Workplace Procrastination." The study explores an important and timely topic, offering valuable insights into authentic leadership's impact on workplace behaviors. However, there are critical methodological concerns that need to be addressed.

Firstly, the inclusion of a qualitative variable such as 'gender' in the correlation matrix is statistically not applicable. This practice is inappropriate as it violates the assumptions of correlation analysis and diminishes the statistical rigor of the study. I strongly recommend revisiting this analysis and excluding qualitative variables from the correlation computations.

Secondly, the justification for employing covariance-based structural equation modeling (CB-SEM) instead of partial least squares structural equation modeling (PLS-SEM) needs to be clarified. Given the nature of the data and the nonparametric advantages of PLS-SEM, it would be a more suitable choice, particularly if the data does not meet the normality assumption required for CB-SEM. The authors should elaborate on the rationale for their methodological decision, including any tests of normality performed to justify the use of CB-SEM.

Addressing these points will significantly enhance the methodological robustness and credibility of the study's findings. Thank you for considering these suggestions, and I look forward to seeing the revised version of this promising work.

Author Response

Response to Reviewer 2 Comments

Thank you for your thoughtful review and the opportunity to address the methodological concerns you have raised regarding our article, "Interpersonal Dynamics of Authentic Leadership: Effects on Support Perception and Workplace Procrastination." We appreciate your recognition of the importance and timeliness of our topic, and we agree that addressing these methodological issues is crucial for enhancing the validity and impact of our study.

We have carefully considered each concern and have undertaken significant revisions to ensure that our methodology is robust and transparent. These adjustments are detailed in the revised manuscript, which we believe now meets the rigorous standards required.

We are grateful for your insights, which have been instrumental in improving our work. We hope that the changes we have made address your concerns satisfactorily and look forward to your further guidance.

Comment 1: Firstly, the inclusion of a qualitative variable such as 'gender' in the correlation matrix is statistically not applicable. This practice is inappropriate as it violates the assumptions of correlation analysis and diminishes the statistical rigor of the study. I strongly recommend revisiting this analysis and excluding qualitative variables from the correlation computations.

Response 1: Thank you for your insightful comments regarding the inclusion of the qualitative variable 'gender' in our correlation matrix. We have taken your feedback into account and have revised Table 3, Descriptive Statistics and Correlations, by excluding 'gender' from the correlation computations to ensure adherence to statistical best practices.

We acknowledge that while the inclusion of qualitative variables like gender in correlation matrices is a common practice in many studies, we appreciate your emphasis on maintaining statistical rigor and the importance of adhering strictly to the assumptions of correlation analysis.

Comment 2: Secondly, the justification for employing covariance-based structural equation modeling (CB-SEM) instead of partial least squares structural equation modeling (PLS-SEM) needs to be clarified. Given the nature of the data and the nonparametric advantages of PLS-SEM, it would be a more suitable choice, particularly if the data does not meet the normality assumption required for CB-SEM. The authors should elaborate on the rationale for their methodological decision, including any tests of normality performed to justify the use of CB-SEM..

Response 2: Thank you for your constructive feedback on our methodological approach regarding the use of covariance-based structural equation modeling (CB-SEM). Following your recommendation and after careful consideration of the nature of our data and analysis requirements, we have switched our approach to partial least squares structural equation modeling (PLS-SEM).

As detailed in the revised section 2.4. Data Analysis, we employed PLS-SEM due to its flexibility with complex models and its appropriateness for our study's small to medium-sized sample, which also does not meet the normality assumptions required for CB-SEM. PLS-SEM is advantageous in our context as it does not require data normality and is robust in handling multiple relationships between latent variables and indicators. We conducted the analyses using SmartPLS v4.0, assessing statistical significance using the bootstrapping method with 5,000 samples.

This revision required substantial time and effort to reanalyze the data, but we are confident that it has enhanced the scientific rigor and appropriateness of our analysis. We have highlighted these changes in red in the revised manuscript.

During the manuscript review process, the following new references have been incorporated (due to the use of a reference manager, they are not highlighted as new text in the manuscript):

Bakker, A. B. & Demerouti, E. (2007). The Job Demands‐Resources model: State of the art. Journal of Managerial Psychology, 22(3), 309–328. https://doi.org/10.1108/02683940710733115

Bandura, A. (1977). Social Learning Theory. Prentice-Hall.

Bandura, A. (1999). Social Cognitive Theory: An agentic perspective. Asian Journal of Social Psychology, 2(1), 21–41. https://doi.org/10.1111/1467-839X.00024

Deci, E. L. & Ryan, R. M. (1985). Toward an organismic integration theory. In Intrinsic motivation and self-determination in human behavior (pp. 113–148). Springer US. https://doi.org/10.1007/978-1-4899-2271-7_5

Deci, E. L. & Ryan, R. M. (2014). Autonomy and need satisfaction in close relationships: Relationships motivation theory. In N. Weinstein (Ed.), Human motivation and interpersonal relationships: Theory, research, and applications (pp. 53–73). Springer.

Hair, J. F., Hult, G. T. M., Ringle, Christian. M. & Sarstedt, M. (2017). A primer on Partial Least Squares Structural Equation Modeling (PLS-SEM) (2nd ed.). SAGE.

Henseler, J., Ringle, C. M. & Sarstedt, M. (2015). A new criterion for assessing discriminant validity in variance-based structural equation modeling. Journal of the Academy of Marketing Science, 43(1), 115–135. https://doi.org/10.1007/s11747-014-0403-8

Meyer, J. P. & Allen, N. J. (1991). A three-component conceptualization of organizational commitment. Human Resource Management Review, 1(1), 61–89. https://doi.org/10.1016/1053-4822(91)90011-Z

Prem, R., Scheel, T. E., Weigelt, O., Hoffmann, K. & Korunka, C. (2018). Procrastination in daily working life: A diary study on within-person processes that link work characteristics to workplace procrastination. Frontiers in Psychology, 9. https://doi.org/10.3389/fpsyg.2018.01087

Regan, S., Laschinger, H. K. S. & Wong, C. A. (2016). The influence of empowerment, authentic leadership, and professional practice environments on nurses’ perceived interprofessional collaboration. Journal of Nursing Management, 24(1), E54–E61. https://doi.org/10.1111/jonm.12288

Ringle, C. M., Wende, S. & Becker, J.-M. (2024). SmartPLS 4. SmartPLS GmbH. http://www.smartpls.com

Spielberger, C. D. & Reheiser, E. C. (2020). Measuring occupational stress: The job stress survey. In R. Crandall (Ed.), Occupational stress (pp. 51–69). CRC Press.

Steiger, J. H. (1990). Structural model evaluation and modification: An interval estimation approach. Multivariate Behavioral Research, 25(2), 173–180. https://doi.org/10.1207/s15327906mbr2502_4

Steinert, C., Heim, N. & Leichsenring, F. (2021). Procrastination, perfectionism, and other work-related mental problems: Prevalence, types, assessment, and treatment—A scoping review. Frontiers in Psychiatry, 12. https://doi.org/10.3389/fpsyt.2021.736776

Tajfel, H. & Turner, J. C. (1979). An integrative theory of intergroup conflict. In W. G. Austin & S. Worchel (Eds.), The social psychology of intergroup relations (pp. 37–47).

Yan, B. & Zhang, X. (2022). What research has been conducted on procrastination? Evidence from a systematical bibliometric analysis. Frontiers in Psychology, 13. https://doi.org/10.3389/fpsyg.2022.809044

Thank you again for your valuable feedback. We appreciate your thorough review and constructive suggestions.

Round 2

Reviewer 1 Report

Dear Authors,

Thank you very much for allowing me to review this updated version of your article. I have carefully read this second version and it has been substantially improved.

The authors improve the abstract based on the various suggestions. The introduction of the article includes the knowledge gap that the authors intend to address and the unique contributions of the study. The relationships between variables are now much clearer. However, the contextualization of the study could be broader. All hypotheses have the necessary theoretical underpinning. Figure 1 is substantially improved.

The authors clarify all doubts about the process and the participants. The ethical justification proposed by the authors is acceptable. Data analysis and results are ostensibly improved, e.g., the authors include convergent and discriminant validity. The validation of the hypotheses is clearer, and the tables improve their content. The discussion of results is improved without being extraordinary. The authors improve the limitations and future research. The authors include a section on theoretical and practical implications that is generally acceptable. Finally, the authors include a section on conclusions.

In general, the article is ready for publication.

Detailed comments on the review are provided above.

Author Response

Dear Reviewer,

We sincerely appreciate your meticulous evaluation and insightful comments regarding the second version of our manuscript: “Interpersonal Dynamics of Authentic Leadership: Effects on Support Perception and Workplace Procrastination”. We are gratified by your positive assessment of the substantial improvements made and your recommendation for publication.

In response to your comment on the study’s contextualization, we have revised and expanded this section to provide a more comprehensive and robust framework, thereby enhancing the applicability and interpretability of our findings. Additionally, we have extended our analyses and provided further detail in the results section, and both the discussion and conclusions have been augmented to offer a more thorough interpretation of the outcomes.

During this revision, we have also incorporated the following new references to further reinforce the theoretical and methodological underpinnings of our study. (Please note that, due to the use of our bibliographic management software, these references have not been updated in color within the text):

  • Fornell, C. & Larcker, D. F. (1981). Evaluating structural equation models with unobservable variables and measurement error. Journal of Marketing Research, 18(1), 39–50. https://doi.org/10.1177/002224378101800104
  • Garrett, R. K. & Danziger, J. N. (2008). On cyberslacking: Workplace status and personal internet use at work. CyberPsychology & Behavior, 11(3), 287–292. https://doi.org/10.1089/cpb.2007.0146
  • Koay, K. Y. & Poon, W. C. (2023). Students’ cyberslacking behaviour in e-learning environments: The role of the Big Five personality traits and situational factors. Journal of Applied Research in Higher Education, 15(2), 521–536. https://doi.org/10.1108/JARHE-11-2021-0437
  • Hair, J. F., Risher, J. J., Sarstedt, M. & Ringle, C. M. (2019). When to use and how to report the results of PLS-SEM. European Business Review, 31(1), 2–24. https://doi.org/10.1108/EBR-11-2018-0203
  • Tandon, A., Kaur, P., Ruparel, N., Islam, J. U. & Dhir, A. (2022). Cyberloafing and cyberslacking in the workplace: Systematic literature review of past achievements and future promises. Internet Research, 32(1), 55–89. https://doi.org/10.1108/INTR-06-2020-0332
  • Venkatesh, V., Cheung, C., Davis, F. & Lee, Z. (2023). Cyberslacking in the workplace: Antecedents and effects on job performance. MIS Quarterly, 47(1), 281–316. https://doi.org/10.25300/MISQ/2022/14985
  • Vitak, J., Crouse, J. & LaRose, R. (2011). Personal Internet use at work: Understanding cyberslacking. Computers in Human Behavior, 27(5), 1751–1759. https://doi.org/10.1016/j.chb.2011.03.002

We once again thank you for your constructive feedback, which has been instrumental in refining our manuscript.

Sincerely,

The Authors

Reviewer 2 Report

Thank you for the opportunity to review the article titled "Interpersonal Dynamics of Authentic Leadership: Effects on Support Perception and Workplace Procrastination."

The article presents a well-structured study on the influence of Authentic Leadership (AL) on workplace procrastination, mediated by perceptions of leader and workgroup support. Below, I provide feedback to enhance the manuscript further:

Graphical Resolution: Improve the resolution of all figures to ensure clarity and professional presentation, especially for Figure 2 and any structural models included.

Tables: In Table 2, it is unnecessary to present the correlation with age, as it adds minimal value to the discussion. Instead, focus on the correlations directly relevant to the study hypotheses and key variables.

Model Assumptions: The manuscript lacks a detailed assessment of model assumptions. It is essential to report measures of:

Internal Consistency Reliability: Confirm composite reliability values (CR) for each construct.

Convergent Validity: Include details on the average variance extracted (AVE) for each latent variable.

Discriminant Validity: Ensure heterotrait-monotrait (HTMT) ratios are reported alongside other metrics like the Fornell-Larcker criterion.

Structural Model Evaluation: The article does not comprehensively evaluate the structural model. Include analyses addressing:

Invariance Testing: Verify measurement invariance across demographic groups (e.g., gender, organization size) if applicable.

Explanatory Power: Highlight R² values for endogenous variables and discuss their adequacy.

Predictive Relevance: Provide Q² values (e.g., using blindfolding procedures) to assess the predictive relevance of the model.

Discussion and Limitations: While the article identifies the absence of mediation effects for cyberslacking, it does not explore this sufficiently. Consider elaborating on potential moderators or alternative mediators (e.g., job complexity or internet accessibility) to explain the non-significant findings. Also, expand on the implications of these limitations for practical interventions.

Future Directions: The study touches briefly on future research areas. Strengthen this section by suggesting:

Investigations into other positive leadership styles (e.g., Servant Leadership) and their relationship with workplace procrastination.

Longitudinal studies to establish causality between AL and procrastination behaviors.

Mixed-method approaches to capture contextual nuances in procrastination.

Ethical Considerations: While ethical compliance is mentioned, explicitly state the institutional ethics committee's approval reference number for added transparency.

In conclusion, the article has a strong foundation and contributes meaningfully to the understanding of Authentic Leadership’s role in mitigating workplace procrastination. Implementing the above improvements will enhance its rigor, clarity, and overall impact. Thank you again for the opportunity to engage with this valuable work.

Comment is in the elaborated considerations.

Author Response

Reviewer 2:

Thank you for the opportunity to review the article titled "Interpersonal Dynamics of Authentic Leadership: Effects on Support Perception and Workplace Procrastination."

The article presents a well-structured study on the influence of Authentic Leadership (AL) on workplace procrastination, mediated by perceptions of leader and workgroup support. Below, I provide feedback to enhance the manuscript further:

We appreciate your detailed comments, which provide interesting and useful insights. Following your recommendations, we have taken different actions to amend the manuscript, trying to answer all the comments suggested.

Graphical Resolution: Improve the resolution of all figures to ensure clarity and professional presentation, especially for Figure 2 and any structural models included.

Thank you for your feedback. We have enhanced the resolution of all figures to ensure greater clarity and a more professional presentation. Additionally, Figure 2 has been updated to incorporate the effects of control variables as requested.

Tables: In Table 2, it is unnecessary to present the correlation with age, as it adds minimal value to the discussion. Instead, focus on the correlations directly relevant to the study hypotheses and key variables.

After careful discussion among the co-authors, we have decided to retain the correlations with age in Table 2. This decision aligns with other revisions, particularly the inclusion of socio-demographic and control variable analyses in the results tables (Table 4). Moreover, we consider age to be a relevant variable, as it demonstrated a small but statistically significant negative effect on soldiering (β = -0.06, p < 0.05). Given this significance, we believe its inclusion contributes to a more comprehensive understanding of the factors influencing procrastination behavior.

Model Assumptions: The manuscript lacks a detailed assessment of model assumptions.

Initially, we analyzed the data using a different structural equation modeling technique. However, in previous review rounds, it was suggested that we employ PLS-SEM due to its methodological advantages. Specifically, PLS-SEM is well-suited for both small and large sample sizes, does not require strict assumptions regarding variable distributions (e.g., normality), and enables the simultaneous analysis of multiple latent variables, each with numerous indicators and hypothesized relationships (Hair et al., 2019). Given these strengths, we adopted PLS-SEM to better align with the recommendations received and to enhance the robustness of our analysis.

It is essential to report measures of:

Internal Consistency Reliability: Confirm composite reliability values (CR) for each construct.

Convergent Validity: Include details on the average variance extracted (AVE) for each latent variable.

Discriminant Validity: Ensure heterotrait-monotrait (HTMT) ratios are reported alongside other metrics like the Fornell-Larcker criterion.

Thank you for bringing this to our attention. In this revised version, we have explicitly reported internal consistency reliability for each study variable, including both Cronbach’s alpha and composite reliability (rho-c), in the Instruments section. Additionally, we have expanded the Results section to clarify that:

"Reliability (Cronbach's alpha and composite reliability) was satisfactory across all scales, with values exceeding .70, as detailed in the Instruments section for each construct."

To address convergent validity, we have now included AVE values in Table 2 and provided further discussion in the paragraph below the table.

Regarding discriminant validity, we now report both the Fornell-Larcker criterion and HTMT ratios to ensure a more rigorous assessment.

The inclusion of these indices has significantly strengthened the manuscript, as all reported values were positive, further confirming the reliability and validity of the measures employed. This improvement enhances the robustness of our findings and provides greater confidence in the measurement model.

Additionally, to facilitate review and highlight these modifications, we have marked all new changes in the manuscript using a different font color.

Structural Model Evaluation: The article does not comprehensively evaluate the structural model. Include analyses addressing:

Invariance Testing: Verify measurement invariance across demographic groups (e.g., gender, organization size) if applicable.

Explanatory Power: Highlight R² values for endogenous variables and discuss their adequacy.

Predictive Relevance: Provide Q² values (e.g., using blindfolding procedures) to assess the predictive relevance of the model.

Thank you for your valuable feedback. In this revised version, we have included R² and Q² values and provided a detailed discussion of these results. Additionally, we have refined our analysis by incorporating four control variables to examine their potential influence more comprehensively. While the relationships between the constructs remain largely unchanged, these modifications enhance the depth of the Results section and strengthen the manuscript.

Regarding demographic group comparisons, we conducted a series of ANOVAs to assess whether significant differences existed in soldiering and cyberslacking behaviors based on gender, educational level, and organization size. The results were as follows:

  • Gender (two levels): No significant differences were found in soldiering behavior, F(1, 736) = 3.74, p = .054, or cyberslacking behavior, F(1, 736) = 0.86, p = .354.
  • Educational level (five levels): No significant differences were found in soldiering behavior, F(4, 733) = 1.41, p = .229. However, significant differences were observed in cyberslacking behavior, F(4, 733) = 3.02, p = .017. Post-hoc analyses revealed that individuals with a basic education level exhibited lower levels of cyberslacking compared to those with higher educational attainment.
  • Organization size (four levels): No significant differences were found in soldiering behavior, F(3, 725) = 0.77, p = .511, or cyberslacking behavior, F(3, 725) = 0.20, p = .895.

These additional analyses provide a more nuanced understanding of the potential influence of demographic factors on the studied behaviors. We appreciate your suggestions, as they have contributed to a more comprehensive and methodologically rigorous manuscript.

Discussion and Limitations: While the article identifies the absence of mediation effects for cyberslacking, it does not explore this sufficiently. Consider elaborating on potential moderators or alternative mediators (e.g., job complexity or internet accessibility) to explain the non-significant findings. Also, expand on the implications of these limitations for practical interventions.

Future Directions: The study touches briefly on future research areas. Strengthen this section by suggesting:

Investigations into other positive leadership styles (e.g., Servant Leadership) and their relationship with workplace procrastination.

Longitudinal studies to establish causality between AL and procrastination behaviors.

Mixed-method approaches to capture contextual nuances in procrastination.

Thank you for your constructive feedback. In response to your suggestions, we have significantly expanded the DiscussionLimitationsFuture Research, and Conclusions sections. The newly added content is highlighted in a different font color for easy identification.

  • Discussion and Limitations: We now provide a more in-depth exploration of the lack of mediation effects for cyberslacking, considering potential moderators (e.g., job complexity, internet accessibility) and alternative mediators that could explain the non-significant findings. Additionally, we have elaborated on the practical implications of these limitations for workplace interventions.
  • Future Directions: We have strengthened this section by incorporating:
    • A discussion on other positive leadership styles (e.g., Servant Leadership) and their potential impact on workplace procrastination.
    • The importance of longitudinal studies to establish causal relationships between authentic leadership and procrastination behaviors.
    • The value of mixed-method approaches to capture contextual nuances in workplace procrastination.

We appreciate your insights, as they have significantly enriched our manuscript. Please refer to the highlighted sections for the specific revisions made.

Ethical Considerations: While ethical compliance is mentioned, explicitly state the institutional ethics committee's approval reference number for added transparency.

We have not included the name of the university where this study was conducted due to the double-blind review process. However, this information has been provided to the editor and, if the manuscript is accepted, it will be fully disclosed in the final published version.

In conclusion, the article has a strong foundation and contributes meaningfully to the understanding of Authentic Leadership’s role in mitigating workplace procrastination. Implementing the above improvements will enhance its rigor, clarity, and overall impact. Thank you again for the opportunity to engage with this valuable work.

Thank you again for your insightful comments and suggestions. We have carefully addressed all concerns raised in this second revision, making substantial improvements to the manuscript.

As part of these revisions, we have incorporated the following new references. Due to the use of a bibliographic manager, they have not been highlighted in color in the text:

  • Fornell, C. & Larcker, D. F. (1981). Evaluating structural equation models with unobservable variables and measurement error. Journal of Marketing Research, 18(1), 39–50. https://doi.org/10.1177/002224378101800104
  • Garrett, R. K. & Danziger, J. N. (2008). On cyberslacking: Workplace status and personal internet use at work. CyberPsychology & Behavior, 11(3), 287–292. https://doi.org/10.1089/cpb.2007.0146
  • Koay, K. Y. & Poon, W. C. (2023). Students’ cyberslacking behaviour in e-learning environments: The role of the Big Five personality traits and situational factors. Journal of Applied Research in Higher Education, 15(2), 521–536. https://doi.org/10.1108/JARHE-11-2021-0437
  • Hair, J. F., Risher, J. J., Sarstedt, M. & Ringle, C. M. (2019). When to use and how to report the results of PLS-SEM. European Business Review, 31(1), 2–24. https://doi.org/10.1108/EBR-11-2018-0203
  • Tandon, A., Kaur, P., Ruparel, N., Islam, J. U. & Dhir, A. (2022). Cyberloafing and cyberslacking in the workplace: Systematic literature review of past achievements and future promises. Internet Research, 32(1), 55–89. https://doi.org/10.1108/INTR-06-2020-0332
  • Venkatesh, V., Cheung, C., Davis, F. & Lee, Z. (2023). Cyberslacking in the workplace: Antecedents and effects on job performance. MIS Quarterly, 47(1), 281–316. https://doi.org/10.25300/MISQ/2022/14985
  • Vitak, J., Crouse, J. & LaRose, R. (2011). Personal Internet use at work: Understanding cyberslacking. Computers in Human Behavior, 27(5), 1751–1759. https://doi.org/10.1016/j.chb.2011.03.002

We trust that our revisions fully address the points raised and that the modifications have significantly strengthened the manuscript. We greatly appreciate the time and effort invested in reviewing our work, and we look forward to your feedback.

Sincerely,

The authors